# Serial Multiple Mediation Analyses: How to Enhance Individual Public Health Emergency Preparedness and Response to Environmental Disasters

**DOI:** 10.3390/ijerph16020223

**Published:** 2019-01-14

**Authors:** Yuxiang Hong, Taesam Lee, Jong-Suk Kim

**Affiliations:** 1School of Management, Hangzhou Dianzi University, Hangzhou 310018, China; hongyx@hdu.edu.cn; 2Department of Civil Engineering, ERI, Gyeongsang National University, 501 Jinju-daero, Jinju, Gyeongnam 660-701, Korea; 3State Key Laboratory of Water Resources and Hydropower Engineering Science, Wuhan University, Wuhan 430072, China

**Keywords:** multiple mediation, public health emergency preparedness, media exposure, knowledge, trust in government

## Abstract

Recent environmental disasters have revealed the government’s limitations in real-time response and mobilization to help the public, especially when disasters occur in large areas at the same time. Therefore, enhancing the ability to prepare for public health emergencies at the grassroots level and extend public health emergency response mechanisms to communities, and even to individual families, is a research question that is of practical significance. This study aimed to investigate mechanisms to determine how media exposure affects individual public health emergency preparedness (PHEP) to environmental disasters; specifically, we examined the mediating role of knowledge and trust in government. The results were as follows: (1) knowledge had a significant mediating effect on the relationship between media exposure and PHEP; (2) trust in government had a significant mediating effect on the relationship between media exposure and PHEP; (3) knowledge and trust in government had significant multiple mediating effects on the relationship between media exposure and PHEP.

## 1. Introduction

Public health emergency preparedness (PHEP) is “the capability of the public health and health care systems, communities, and individuals, to prevent, protect against, quickly respond to, and recover from health emergencies, particularly those whose scale, timing, or unpredictability threatens to overwhelm routine capabilities” [1]. Strengthening emergency preparedness is the primary task of emergency management at the national and local levels [2,3]. However, recent environmental disasters have revealed the limitation of government in real-time response and mobilization to help the public, especially when a disaster strikes a large area at the same time [4]. Public health emergency preparedness is a “bottom-up” system [1], and individuals, families, and communities, industries are essential to increasing the resilience of emergency management [5,6,7]. They are the first respondents coping with disasters before the arriving of support from government. As responding at the first time and the first scene plays important roles on the mitigation of disaster losses, how to strengthen PHEP at the grassroots level, and how to extend public health emergency response mechanisms to communities and even families, is a research question of highly practical significance. In this study, we aim to describe the relationship between media exposure and individual PHEP behaviors.

In the Law of the People’s Republic of China on Emergency Response, prevention and emergency preparedness is a separate chapter (No. 7), and media has been given an important responsibility to strengthen emergency preparedness. Media is often regarded as the ‘fourth power’ in addition to the legislative, judicial and administrative powers, and it is exceptionally crucial for promoting emergency preparedness [8,9,10]. Murphy et al. found that watching national news, reading a newspaper, and listening to the radio can positively affect preparedness [11]. Paek et al., found that people who were more exposed to the news were more prepared for various emergencies reported by television, radio, newspapers, and the Internet [9]. However, the existing literature ignored the mediating effects (e.g., knowledge, trust in government) on the relationship between media exposure and emergency preparedness behaviors.

Media exposure can increase an individual’s knowledge and trust in their government. First, media can help people learn about their immediate environment and potential threats therein, and individuals can also be warned by the media in case of imminent danger by creating awareness among them [12]. Second, by setting and initiating the agenda, a culture of public responsibility and safety can be shaped and security awareness can be heightened [13]. Third, media can expand the boundaries of personal experience to enable the public to become aware of various emerging man-made and natural disasters in the world through reporting on disaster-related news and documentaries [14]; this can be supported by Social Learning Theory [15]. In addition, media can help people learn about relevant government processes, activities, and government’s performance, especially the outcome of the government’s response to improve the public’s sense of transparency, and thus increase their trust in government [16,17].

People’s behavior is strongly affected by confidence in their ability to perform [18]; thus, knowledge of disasters and government activities can increase their perceived behavioral control, thereby improving their PHEP. Trust has been shown to have a positive effect on a person’s degree of cooperative behavior [19,20,21,22,23]. DeCremer et al., found that trust in government has been shown to affect a person’s degree of cooperative behavior [24,25]. Lasker found that trust in government can lead a person to become more cooperative with the government in an actual emergency [26]. Based on a survey of 316 drill participants, Allen found that trust-building between local government and community members can promote emergency preparedness [27]. Besides, during public health emergencies, the more the people have knowledge of the nature and evolution of disasters as well as a basic understanding of the government’s response capability and rules, the more the people trust in the government’s capability [16,17]. 

Thus, we propose the following hypothesis: knowledge and trust in government has significant serial multiple mediating effects on the relationship between media exposure and individual public health preparedness behaviors to environmental disasters.

## 2. Materials and Methods

### 2.1. Study Participants

In China, after the outbreak of Severe Acute Respiratory Syndrome (SARS) in 2003, more attention had been paid to government information disclosure. Traditional media (e.g., newspaper, magazine, radio, TV) were effectively used by the government to disclose public information, such as government performance and public service information. In 2006, the national government website system was established, from when government information disclosure can be carried out through new media (e.g., Internet, mobile phone customization). In late 2013, the central government of China put Micro. blog and WeChat as equally important new media channels of government information disclosure as government websites.

Based on a review of the website maintained by the Health and Family Planning Commission of Hangzhou Municipality, which is one of the departments responsible for public health, we found that Hangzhou began releasing official public health information on Micro. blog and WeChat in 2014 but began disclosing official public health information much earlier. 

As shown in Figure 1, we found that official public health information disclosure on government websites is relatively flat every year. This was true even in 2013 during the Avian influenza A (H7N9) outbreak, when information disclosure on traditional media peaked. In 2014, the government began to use WeChat and Micro. blog to release information. We found that from 2014 to 2016, official public health information disclosure on WeChat and Micro. blog was gradually increasing, while information disclosure on traditional media was steadily decreasing and remaining flat on government websites. Thus, we can predict that WeChat and Micro. blog were gradually supplanting traditional media. Based on this observation, we believe that both traditional and new media played a role in public health information disclosure in Hangzhou, and their purposes are continuously changing, which is representative of all of China. As such, we chose samples from Hangzhou. 

This study was based on a household survey conducted in Hangzhou City in Zhejiang Province, China (Figure 2), from July to September 2015. Hangzhou was one of the first cities in China using new media to release official information. A total of 800 questionnaires were distributed through door-to-door visits to households in Hangzhou. A total of 747 questionnaires were collected and 702 were included in analysis, as 45 questionnaires were discarded due to missing data. The socio-demographic information for participants is summarized in Table 1.

### 2.2. Measures

Referring to the questionnaires used by Fleming et al. [28] and by Murphy et al. [11], our six media exposure measures included ‘read a newspaper/read a magazine/listen to the radio/watch TV/use the Internet/use mobile phone customization.’; used a 5-point Likert-type scale ranging from 1 (never) to 5 (very often). Specifically, ‘read a newspaper/read a magazine/listen to the radio/watch TV’ represented traditional media and ‘use the Internet/use mobile phone customization’ represented new media. Cronbach’s alpha was calculated for the traditional media exposure scale (α = 0.73), as well as for the new media exposure scale (α = 0.62).We measured knowledge by asking respondents to estimate the degree to which they knew about four types of public health, including infectious diseases, food security, air pollution, and drinking water pollution, using a 5-point Likert-type scale from 1 (know nothing) to 5 (know a lot). Cronbach’s alpha was calculated for the scale (α = 0.90). We measured trust in government by asking respondents to estimate the degree to which they trust the government’s capability to respond to public health emergencies on a5-point Likert-type scale from 1 (do not trust at all) to 5 (trust very much).

In this study, we divided preparedness behavior into cooperation behavior and supplies [29,30]. Cooperative behavior (COOP) refers to public health emergency behavior that requires the collaboration of two or more individuals and agents; e.g., skills training, emergency drill, volunteer activity. Supplies (SUP) is one kind of non-cooperative behavior that can be implemented by one individual. Referring to the questionnaires used by Murphy et al. [11] and Paek et al. [9] in order to measure cooperation behaviors, we asked respondents to estimate the frequency of engaging in the following three community interactions on a 5-point Likert-type scale of 1 (never) to 5 (very often): attend emergency skills training, attend emergency drills, and attend emergency volunteer service. Cronbach’s alpha was calculated for the scale (α = 0. 91). To measure behaviors regarding emergency supplies, we asked individuals to indicate which of the following three recommended items they owned: drinking water and emergency food for 72 h, a first aid kit, and a cloth face mask. The scores for emergency supplies ranged from 0 (owned none of these) to 3 (owned all of the recommended supplies). Since behaviors of cooperation and emergency supply ownership are two independent dimensions, we measured emergency preparedness behaviors by averaging them after normalization. 

Gender, age, education, and household income were selected as control variables, as previous studies have verified that they can significantly affect emergency preparedness behaviors [9,11,31,32,33,34]. Figure 3 shows the preliminary analysis process, and the hypothesis verification and evaluation procedures through multiple regression models used in this study.

### 2.3. Intentionally Biased Bootstrapping Method

Bootstrapping analysis is a statistical method for generating replicated data sets from observed data and evaluating the variability of the quantiles of interest without analytical calculation [35,36]. An intentionally biased bootstrapping (IBB) technique is the resampling of explanatory variables with the response variables while increasing or decreasing the mean of the explanatory variable by a certain level. From this IBB, the effect of the response variable can be assessed by changing the mean [36,37]. The mathematical description of the IBB method applied in this study is as follows.

Suppose that, among *n* observations $x_{i}$, where i=1, 2, 3, …, n, we resample the observations with bootstrapping by increasing the mean of the simulated data up to Δμ. This IBB can be done by assigning different weights $S_{i,n}$ regarding the order of the observations as follows (Equation 1):(1)$$S_{i,n}=i/n$$

The assigned weight $S_{i,n}$ is the selected probability for the observations after scaling and adjustment. The average of the resampled data is shown in Equation (2): (2)$$\tilde{\mu }=\frac{1}{\psi }\sum_{i=1}^{n}S_{i,n}x_{\left(i\right)},$$
where $x_{\left(i\right)}$ represents the i-th ordered value and $\psi =\sum_{i=1}^{n}S_{i,n}$. The amount of the average increase in ∆μ is:(3)$$\Delta \mu =\frac{1}{\psi }\sum_{i=1}^{n}S_{i,n}x_{\left(i\right)}-\frac{1}{n}\sum_{i=1}^{n}x_{i}$$

In order to obtain another value $\Delta \tilde{\mu }\left(r\right)$, the weights can be generalized as: (4)$$\Delta \tilde{\mu }\left(r\right)=\tilde{\mu }\left(r\right)-\hat{\mu }=\frac{1}{\psi_{r}}\sum_{i=1}^{n}s_{i,n}^{r}x_{\left(i\right)}-\frac{1}{n}\sum_{i=1}^{n}x_{i}$$
where $\psi_{r}=\sum_{i=1}^{n}s_{i,n}^{r}$ and $\tilde{\mu }\left(r\right)=\frac{1}{\psi_{r}}\sum_{i=1}^{n}s_{i,n}^{r}x_{\left(i\right)}$. If the value of the average increase (or decrease) is given as Δμ, the weight order “r” is calculated accordingly. The selection of the weight order (*r*) can be performed using an optimization technique with the objective function of minimize ${\left[\Delta \mu -\Delta \tilde{\mu }\left(r\right)\right]}^{2}$.

In this study, the IBB technique was employed to generate resampled data sets for PHEP and response to environmental disasters.

### 2.4. Serial Multiple Mediator Model

The Multiple Mediator Model in general has two forms that can be divided by whether the mediators are linked together (the serial multiple mediator models) or not (the parallel multiple mediator models) [38]. The model applied in this study is a serial multiple mediator model, which has three indirect effects and one direct effect. The three indirect paths are found by tracing all possible ways of getting from X to Y through at least one M: X → M_1_→ Y;
X → M_2_ → Y;
X → M_1_ → M_2_ → Y.

Two serial multiple mediator models can translate into three equations:M_1_ = β_01_ + β_1_X + ε_1_;(5)
M_2_ = β_02_ + β_2_M_1_ + β_5_X + ε_2_;(6)
Y = β_03_ + β_4_X + β_3_M_2_ + β_6_M_1_ + ε_3_:(7)

Therefore, the indirect effect of X on Y through M_1_ is β_1_β_6_, the indirect effect through Model 2 is β_5_β_3_, the indirect effect through M_1_ and M_2_ in serial is β_1_β_2_β_3_, and the total indirect effect of X is β_1_β_6_+β_5_β_3_+β_1_β_2_β_3_. Taylor et al. (2008) compared different categories of methods for testing mediation and found the bootstrap methods were the best performers [39]. In this study, we also used the bootstrap method to test the serial multiple mediating effects.

## 3. Results

### 3.1. Preliminary Analysis

We conducted a confirmatory factor analysis (CFA) on six variables including traditional media exposure, new media exposure, knowledge, trust in government, cooperation behaviors, and supply behaviors. The result showed that the Kaiser-Meyer-Olkin value was 0.83 > 0.7, and Bartlett’s Test of Sphericity was significant. This result means that the data has good structural validity, and is suitable for factor analysis. Then, we used the Amos 17.0 software program (SPSS Inc., Chicago, IL, USA) to verify the discriminant validity of variables in this study. We included all six variables in a measurement model and made freely covarying latent constructs. The results indicate that the six variables had proper discriminant validation that can represent six different constructs (Table 2). Means, standard deviations, and the correlation coefficients among the variables are reported in Table 3. These results provide initial support for the hypotheses in this study.

### 3.2. Serial Multiple Mediation Analysis

To maintain the data in the same dimension, we min-max normalized the data. Regression analysis was first used to test the study’s hypothesis by controlling for gender, age, education, and household income. According to the regression results presented in Table 4, media exposure had significant positive effects on PHEP (β = 0.202, *p* < 0.001), on knowledge (β = 0.363, *p* < 0.001), and on trust in government (β = 0.111, *p* < 0.05). Knowledge had significant positive effects on PHEP (β = 0.240, *p* < 0.001) and on trust in government (β = 0.286, *p* < 0.001). Trust in government had significant positive effects on PHEP (β = 0.195, *p* < 0.001).

Then, we used Model 6 from PROCESS [38]. We estimated 5000 bootstrap samples in which the independent variable was media exposure, the mediators were knowledge and trust in government, and the dependent variable was PHEP. We also included gender, age, education, and household income as covariates in the model. The results indicated that knowledge and trust in government partly mediate the relationship between media exposure and PHEP (total indirect effect = 0.1290; 95% CI: [0.0884, 0.1814]; direct effect = 0.2019, 95% CI: [0.1006, 0.3032]; see Figure 4). Specifically, knowledge can mediate the relationship between media exposure and PHEP (indirect effect=0.0870; 95% CI: [0.0538,0.1315]); trust in government can mediate the relationship between media exposure and PHEP (indirect effect = 0.0217; 95% CI: [0.0050, 0.0488]); and the multiple mediation effects of knowledge and trust in government on the relationship between media exposure and PHEP was significant (indirect effect = 0.0203; 95% CI: [0.0100, 0.0354]). Therefore, the hypothesis was supported. 

To more specifically understand the mediating effect of knowledge and trust in government on the relationship between media exposure and public health emergency preparedness behaviors, we built four scenarios with two types of media exposure and two types of PHEP. In addition, the effects of the two types of PHEP were analyzed using the intentionally biased bootstrapping (IBB) model (Figure 5). For the media exposure-cooperation (ME-COOP) behavior scenarios (Scenarios 1 and 3), the most sensitive variable to change with COOP was government trust (GT), followed by knowledge (KN) and ME. On the other hand, ME and emergency supply behaviors (SUP) in Scenarios 2 and 4 showed relatively little change compared to those in ME-COOP. The most sensitive factor to changes in SUP was KN. The impact of GT was found to be relatively minor. New media exposure (NME) appeared to be larger than changes in traditional media exposure (TME). 

In the TME-COOP scenario, results indicated that knowledge and trust in government can completely mediate the relationship between traditional media exposure and cooperation behaviors (total indirect effect = 0.1142, 95% CI: [0.0719, 0.1653]; direct effect = 0.0552, 95% CI: [−0.0292, 0.1395]; see Scenario 1 in Figure 5). Specifically, knowledge can mediate the relationship between traditional media exposure and cooperation behaviors (indirect effect = 0.0508; 95% CI: [0.0229, 0.0848]);trust in government can mediate the relationship between traditional media exposure and cooperation behaviors (indirect effect = 0.0344; 95% CI: [0.0083, 0.0692]); and the multiple mediation effects of knowledge and trust in government on the relationship between traditional media exposure and cooperation behaviors was significant (indirect effect = 0.0289; 95% CI: [0.0165, 0.0463]). 

In the TME-SUP scenario, results indicated that knowledge and trust in government can partly mediate the relationship between traditional media exposure and emergency supply ownership behaviors (total indirect effect = 0.1126, 95% CI: [0.0736,0.1850]; direct effect = 0.1764, 95% CI: [0.0209, 0.3318]; see Scenario 2 in Figure 5). Specifically, knowledge can mediate the relationship between traditional media exposure and emergency supply ownership behaviors (indirect effect = 0.1086; 95% CI: [0.0616, 0.1705]), while the mediating effect of trust in government on the relationship between traditional media exposure and emergency supply ownership behaviors was not significant (indirect effect = 0.0076; 95% CI: [−0.0045, 0.0302]). Moreover, the multiple mediation effects of knowledge and trust in government on the relationship between traditional media exposure and cooperation behaviors was not significant (indirect effect = 0.0064; 95% CI: [−0.0049, 0.0209]). 

In the NME-COOP scenario, results indicated that knowledge and trust in government can completely mediate the relationship between new media exposure and cooperation behaviors (total indirect effect = 0.1126, 95% CI: [0.0736,0.1850]; direct effect = 0.0459, 95% CI: [−0.0175, 0.1094]); see Scenario 3 in Figure 5). Specifically, knowledge can mediate the relationship between traditional media exposure and cooperation behaviors (indirect effect = 0.0291; 95% CI: [0.0129, 0.0517]), and the multiple mediation effects of knowledge and trust in government on the relationship between new media exposure and cooperation behaviors was significant (indirect effect = 0.0173; 95% CI: [0.0102, 0.0293]). However, the mediating effect of trust in government on the relationship between new media exposure and cooperation behaviors was not significant (indirect effect = 0.0142; 95% CI: [−0.0074, 0.0367]). 

In the NME-SUP scenario, results indicated that knowledge and trust in government can partially mediate the relationship between new media exposure and emergency supply ownership behaviors (total indirect effect = 0.0678, 95% CI: [0.0375, 0.1074]; direct effect = 0.1952, 95% CI: [0.0787, 0.3116]; see Scenario 4 in Figure 5). Specifically, knowledge can mediate the relationship between new media exposure and emergency supply ownership behaviors (indirect effect = 0.0606; 95% CI: 0.0326, 0.1004)), while the mediating effect of trust in government on the relationship between new media exposure and emergency supply ownership behaviors was not significant (indirect effect = 0.0033; 95% CI: [−0.0021, 0.0182]). Further, the multiple mediation effects of knowledge and trust in government on the relationship between new media exposure and emergency supply ownership behaviors was not significant (indirect effect = 0.0040; 95% CI: [−0.0028, 0.0129]). 

## 4. Discussion

Based on this study, there is the reason to believe that increasing an individual’s media exposure can increase their PHEP. Specifically, frequent exposure to both traditional media and new media can directly promote their ownership of public health supplies behavior. Although the direct effects of the relationship between media exposure and their cooperation behaviors was not significant, knowledge and trust in government mediated the relationship between media exposure and cooperation behaviors. Although different opinions exist related to the effect of media exposure on trust in government, the media malaise thesis indicates that mass media has an adverse effect on trust [40], while the mobilization thesis contends that mass media has a positive effect on trust [41], and the neutral thesis suggests that the relationship between media exposure and political trust is highly dependent on existing conditions [42]. Regardless of which research articles prevail, in the Chinese context, media is always a significant way to strengthen political trust; this positive effect has been empirically verified in our study and other academic research [43]. However, in the cross-scenario analysis, we found that trust in government can play roles in the relationship between media exposure and cooperation behaviors, while it did not play roles in the relationship between media exposure and supplies behaviors. Moreover, new media exposure does not have direct effects on trust in government in the NME-COOP scenario, which means that simple media exposure may not increase individuals’ trust in government, but needs to through the growth in knowledge. Therefore, through the effective combination of new and traditional media platforms, various forms of public safety contents for distributing knowledge of public health emergencies, public health emergency management, and government activities and events should be communicated to the public in a seamless, comprehensive, and real-time manner in order to improve their trust in government.

Trust in government can increase emergency cooperation behaviors [24,25,26], and this was tested again in the context of PHEP. To improve PHEP cooperation behaviors, the level of trust in government should be increased, and the government should build good relationships with the community and the public. The executors and planners of emergency and preparedness need to be part of the community. They need to regularly meet, listen to, and become familiar with community representatives and residents. The content of these meetings may be public health information of interest to residents as well as emergency preparedness topics. Knowledge is a significant predictor for both types of PHEP, so disaster-related education programs should be developed and promoted throughout the community. The content and form of programs should be well-designed and based on the particular learning needs of different target groups, such as the ‘trickle-down’ process of dissemination for children. Community groups can be established for women, and special education can be provided for elderly and disabled people [44]. Because trust in government can promote cooperation behaviors, programs for education need to include an introduction to disaster-related government rules, procedures, actions, and events.

Furthermore, even though traditional media exposure can still affect PHEP, the younger and more highly educated people tend to use new media. Due to the advent of new media, the government cannot completely control media, and the new media users maintain a cautious and conservative attitude towards cooperation behaviors. New media exposure cannot directly lead to cooperation but must do so through an understanding and trust in government. Therefore, in order to promote PHEP through new media, simple verbal communication is ineffective, and more useful knowledge and skills should be disseminated. This requires not only the creation and cooperation of new media content providers, but the government also needs to participate in dissemination through new media. In fact, new media is an important way for the government to communicate with the public and build trust in government. Through new media, the government can give more feedback to the public’s inquiries and needs, as well as provide more channels for public participation and help the public to better understand and support the government’s emergency management activities.

## 5. Conclusions

This study analyzed the influence of media exposure on public health emergency preparedness behaviors to environmental disasters. Previous studies have conducted extensive empirical research on this topic; however, there has been relatively little study of the mechanisms with which media exposure impacts PHEP, as well as little consideration of the mediating effects of knowledge and trust in government. In this study, we constructed a model with influence mechanisms and performed an empirical analysis. The study’s hypothesis of serial multiple mediating effects was supported. First, this study confirmed that media exposure could significantly and positively affect knowledge, trust in government, and PHEP. Knowledge had a significant and positive effect on trust in government and PHEP. Trust in government had a significant and positive effect on PHEP. Second, we found that knowledge and trust in government can individually mediate the relationship between media exposure and PHEP. Third, the empirical study also verified that knowledge and trust in government have serial multiple mediating effects on the relationship between media exposure and PHEP. Last, we found that the mediating effects are different among the four scenarios that combined two types of media exposure and two types of PHEP. The effect of media exposure on public health emergency preparedness was considered an important research question. In this study, we divided media exposure into traditional media exposure and new media exposure, and divided public health emergency preparedness into cooperation behavior and supplies. Through the role of knowledge and trust in government as mediators, we constructed a serial multiple mediation model with four scenarios. We suggest that future research could build on the new analysis framework of this study, expand the scope of the samples, and carry out some cross-cultural studies, thus revealing more meaningful conclusions.

## Figures and Tables

**Figure 1 ijerph-16-00223-f001:**
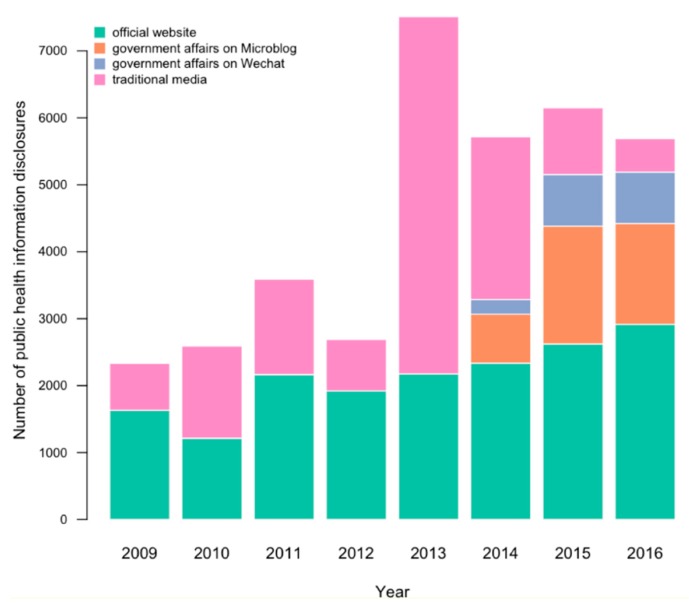
Disclosure of public health information in Hangzhou City, China.

**Figure 2 ijerph-16-00223-f002:**
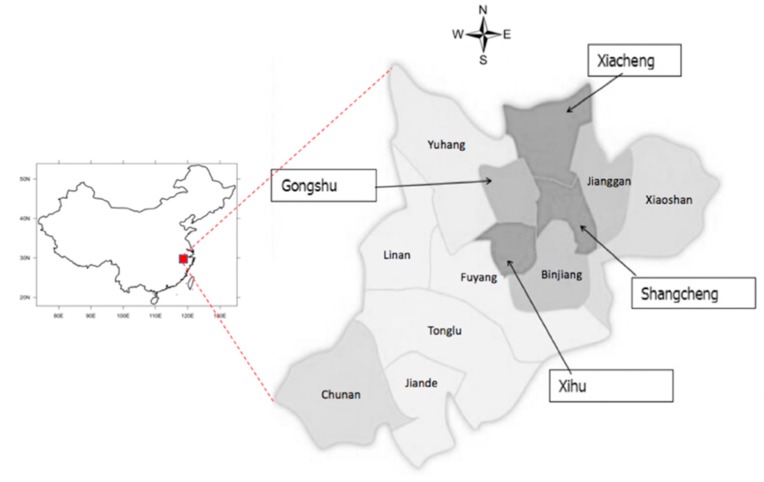
Location of the study area. The study was conducted in four districts (Gongshu, Xiacheng, Shangcheng, and Xihu) of Hangzhou, China.

**Figure 3 ijerph-16-00223-f003:**
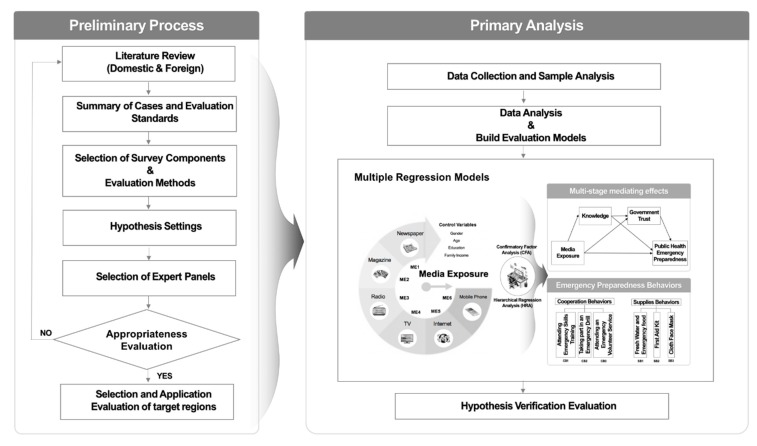
Preliminary analysis process and hypothesis verification procedures of this study.

**Figure 4 ijerph-16-00223-f004:**
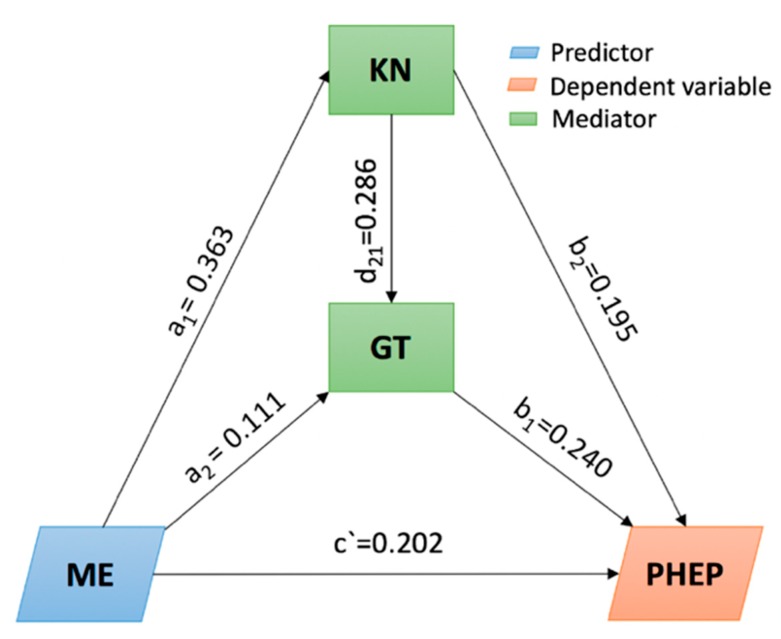
Serial multiple mediating effects of knowledge (KN) and government trust (GT) on the relationship between media exposure (ME) and public health emergency preparedness (PHEP).

**Figure 5 ijerph-16-00223-f005:**
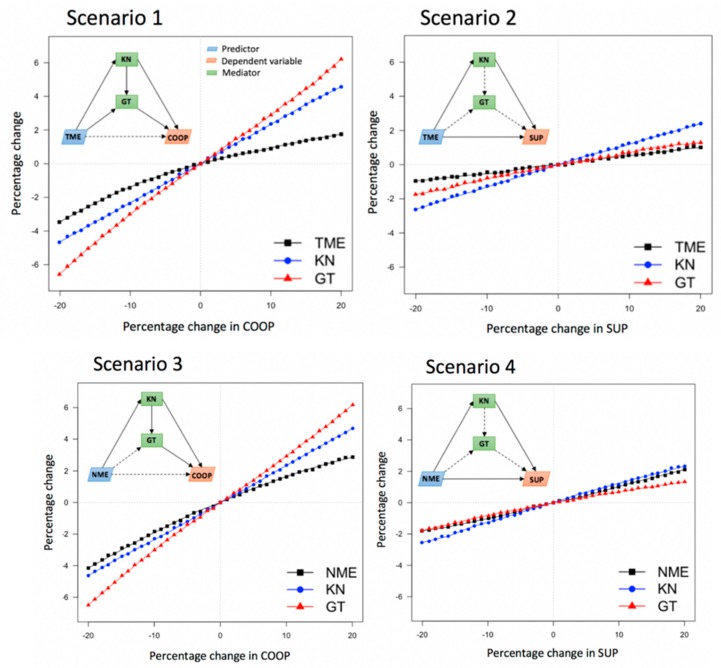
Mediating effects of four different scenarios. Percentage changes in each component, such as TME, NME, KN, and GT, were shown for the percentage changes in PHEP (COOP and SUP) from −20% to +20% with 1% intervals using the Intentionally Biased Bootstrapping (IBB) model.

**Table 1 ijerph-16-00223-t001:** Descriptive analysis of socio-demographic characteristics (*N* = 702).

Variables	Socio-Demographics	Frequency	Percentage
Gender	males	342	48.7
	females	360	51.3
Age	20 years old and younger	21	3.0
	21 to 30 years old	177	25.2
	31 to 40 years old	188	26.8
	40 to 50 years old	162	23.0
	50 to 60 years old	119	17.0
	older than 60 years old	35	5.0
Education	less than high school	89	12.7
	high school degree	131	18.7
	junior college degree	144	20.5
	bachelor’s degree	305	43.4
	master’s degree or higher	33	4.7
Annual family income	lower than CNY 80,000 (USD 11,550)	241	34.3
	CNY 80,000–120,000 (USD 11,550–17,325)	188	26.8
	CNY 120,000–200,000 (USD 17,325–28,876)	174	24.8
	CNY 200,000–300,000 (USD 28,876–43,313)	70	10.0
	higher than CNY 300,000 (USD 43,313)	29	4.1

**Table 2 ijerph-16-00223-t002:** Results of the confirmatory factor analysis (CFA).

Models	χ^2^/df	TLI	CFI	RMSEA
Six-factor model: TME; NME; KN; GT; COOP; SUP	2.748	0.952	0.963	0.048
Five-factor model: TME; NME; KN; GT; COOP; SUP	3.505	0.932	0.945	0.058
Five-factor model: TME+NME; KN; GT; COOP; SUP	3.953	0.920	0.935	0.063
Four-factor model: TME+NME; KN; GT; COOP; SUP	4.624	0.901	0.917	0.070
Three-factor model: TME+NME; KN+GT; COOP; SUP	5.177	0.886	0.903	0.075
Two-factor model: TME+NME+KN+GT; COOP; SUP	9.403	0.772	0.802	0.106
Singer-factor model: TME+NME+KN+GT+COOP+ SUP	21.687	0.438	0.508	0.167

Note. TLI = Tucker-Lewis Index, CFI = Comparative Fit Index, RMSEA = Root Mean Square Error of Approximation, TME = Traditional media exposure, NME = New media exposure, KN = Knowledge, GT = government trust, COOP = Cooperation behaviors, SUP = Supplies behaviors.

**Table 3 ijerph-16-00223-t003:** Means, standard deviations, and correlation coefficients.

Variables	Means	SD	1	2	3	4	5	6	7	8	9
1 TME	2.99	0.78									
2 NME	3.42	1.07	0.17 *								
3 KN	3.26	0.70	0.31 *	0.20 *							
4 GT	3.28	0.71	0.17 *	0.10 **	0.30 *						
5 COOP	3.22	0.75	0.14 *	0.13 *	0.27 *	0.36 *					
6 SUP	1.56	0.99	0.16 *	0.19 *	0.24 *	0.11 *	0.17 *				
7 GE	1.51	0.50	−0.01	0.10 *	−0.01	−0.02	−0.00	0.08 **			
8 AG	3.41	1.25	0.18 *	−0.38 *	0.06	−0.02	−0.02	0.01	0.02		
9 EDU	3.09	1.15	0.01	0.42 *	0.11 *	−0.04	0.08 **	0.09 **	0.14 *	−0.41 *	
10 FINS	3.18	1.23	0.14 *	0.24 *	0.11 *	0.04	0.03	0.14 *	0.08 **	−0.06	0.38 *

Note. **p* < 0.01, ** *p* < 0.05. GE = Gender, 1 = men, 2 = women; AG = Age, 1 = 20 years and younger, 2 = 21 to 30 years, 3 = 31 to 40 years, 4 = 40 to 50 years, 5 = 50 to 60 years, 6 = older than 60; years EDU = Education, 1 = middle school and lower, 2 = high school, 3 = junior college, 4 = bachelor’s degree, 5 = master’s degree or higher. Family Income (RMB per year), 1 = lower than CNY 80,000, 2 = CNY 80,000 to 120,000, 3 = CNY 120,000 to 200,000, 4 = CNY 200,000 to 300,000, 5 = more than CNY 300,000.

**Table 4 ijerph-16-00223-t004:** Results of regression analysis.

Dependent Variables	Independent Variables	*R*	*R* ^2^	*F*	β	*T*
KN	ME	0.345	0.119	18.750 *	0.363	8.481 *
GT	ME	0.328	0.108	13.995 *	0.111	2.411 **
	KN				0.286	7.361 *
PHEP	ME	0.402	0.161	19.056 *	0.202	4.614 *
	KN				0.240	5.328 *
	GT				0.195	4.614 *

Note. **p* < 0.01, ** *p* < 0.05.

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
