# Peer review of "Serial Multiple Mediation Analyses: How to Enhance Individual Public Health Emergency Preparedness and Response to Environmental Disasters"

_ijerph, 2019, doi:10.3390/ijerph16020223_

Reviewer 1 Report

While it's a good approach to look at the bottom up, this paper emphasizes the need to trust government in order to increase cooperation. What is missing in the background is a discussion on the limitations of government in an emergency, where government support will or will not be available, will populations be "on their own" and for how long. For example, to avoid traffic jams and for other reasons "staying in place" may be a best solution.

If people understand that in certain instances the government will not be 'available' to assist them, and why, that is also important.

Thirdly the role of industry has not been addressed, and oftimes an emergency can be industry based, i.e. release of chemicals, natechs - natural and technology disaster (e.g. combined effect of an earthquake and subsequent release of chemicals/energy from industry or nuclear power plant). There is literature and government papers/policies on the need for partnerships between government, communities and industries in mitigating the impact of disasters; especially the need for transparency on what could happen, how to best warn a population and how they can protect themselves.

This paper needs to provide a more comprehensive picture of the problem and present and discuss its limitations, trusting government does not mean the population is better prepared, rather other factors need to be accounted for such as the role of government/industry in supplying best information  so the population can best prepare.

With regard to the text below Figure 1, it may be best to supply this info in a table. If not, the information should be presented more succintly as now each item is 3-4  numbers (N,%, range), (N, %, level of education), income (N, %, range in local currency and again in dollars) etc. Too much info for each sentance. Also, high school certificate (not degree), Junior college is usually also a certificate not degree so participants have certificates (not certificate), and then refer to undergraduate and graduate degrees (the first degree could be a bachelor's but can also be a type of professional degree not called a bachelor).

When you discuss 'measures'  and the questionaire, you should refer to the type of questions (e.g. Likert) and why you chose these variables. Here again you could discuss concepts such as validity and reliability.

Unfortunately I am less familiar with your statistical method and can't evaluate it.

Can your results be interpreted as the more a citizen in  is accustomed to following government directives, the more likely they are to comply if they get exposed to sufficient information, instead of the more information exposure they receive from the media, the more likely they will trust government and comply. I know I'm simplifying here, but am basically asking if this research can transfer to other cultures/populations? if not, the potential difference should be discussed.

Finally, in your discussion you state media exposure can increase PHEP, I suggest more details, i.e. number of media you are exposed to, frequency you view media, if exposure to more than one type of media makes a difference, etc.

Author Response

We are pleased to have an opportunity to revise our paper and sincerely appreciate your constructive and insightful comments, which have improved the manuscript. Below we have replied to the reviewer’s comments and provided extensive explanations for the changes that were made in the revised manuscript. In addition, the revised manuscript was reviewed by professional English editors. We believe the paper has been greatly improved as a result of your feedback and represents a greater contribution to the literature.

Point 1: While it's a good approach to look at the bottom up, this paper emphasizes the need to trust government in order to increase cooperation. What is missing in the background is a discussion on the limitations of government in an emergency, where government support will or will not be available, will populations be "on their own" and for how long. For example, to avoid traffic jams and for other reasons "staying in place" may be a best solution. If people understand that in certain instances the government will not be 'available' to assist them, and why, that is also important. Thirdly the role of industry has not been addressed, and oftimes an emergency can be industry based, i.e. release of chemicals, natechs - natural and technology disaster (e.g. combined effect of an earthquake and subsequent release of chemicals/energy from industry or nuclear power plant). There is literature and government papers/policies on the need for partnerships between government, communities and industries in mitigating the impact of disasters; especially the need for transparency on what could happen, how to best warn a population and how they can protect themselves.

Response 1: Thank you very much for your kind consideration and suggestions.

In order to explain where government support will or will not be available, we need to include the following and describe individuals, families, communities and industries by stating that they “are the first respondents coping with disasters before the arrival of support teams from the government. Due to the fact that the parties involved in responding first at the scene are vitally important, we need to ensure that public health emergency procedures and teams are strengthened at inception and grassroots level, and extending public health emergency response mechanisms to communities and families which is of course, a highly significant.” Besides this consideration, we must consider the role of the industry in defining PHEP. Moreover, since in this study we focus on the topic of “Individual public health emergency preparedness behaviors”, we modified our title to emphasize the topic and added on sentence: “In this study, we aim to find the mechanism that how media exposure influence on the individual public health emergency preparedness behaviors.

Point 2: This paper needs to provide a more comprehensive picture of the problem and present and discuss its limitations, trusting government does not mean the population is better prepared, rather other factors need to be accounted for such as the role of government/industry in supplying best information so the population can best prepare.

Response 2: Thank you very much for your suggestions. By reviewing the existing literature study on the relationship between media exposure and emergency preparedness behaviors, we found that “the existing literature ignored the mediating effects (e.g. knowledge, trust in government) on the relationship between media exposure and emergency preparedness behaviours.” We have described the role of trust in the government in a part of the theoretical analysis as well as in discussion. The empirical results indicated that trust in government cannot predict the supplies behaviors, and in the scenarios of cooperation behavior, the role of trust in government often has been mediated by knowledge, especially in the NME-COOP scenario.

Point 3: With regard to the text below Figure 1, it may be best to supply this info in a table. If not, the information should be presented more succintly as now each item is 3-4 numbers (N, %, range), (N, %, level of education), income (N, %, range in local currency and again in dollars) etc. Too much info for each sentence. Also, high school certificate (not degree), Junior college is usually also a certificate not degree so participants have certificates (not certificate), and then refer to undergraduate and graduate degrees (the first degree could be a bachelor's but can also be a type of professional degree not called a bachelor).

Response 3: Thank you very much for your kind feedback. In the revised manuscript, we have added a table to better describe the socio-demographic characteristics.

Point 4: When you discuss ‘measures’ and the questionnaire, you should refer to the type of questions (e.g. Likert) and why you chose these variables. Here again you could discuss concepts such as validity and reliability.

Response 4: Thank you for bringing our attention to this issue. We have added the note of “using a 5-point Likert-type scale” in some measures. We also gave more details to explain some variables. The validity and reliability tests were modified. In order to test the reliability of the measures, we calculated the Cronbach’s alpha for the scales except the one-item scales. In order to test the validity of the measures, we conducted CFA on six variables including traditional media exposure, new media exposure, knowledge, trust in government, cooperation behaviours and supply behaviours. The result shows that the Kaiser-Meyer-Olkin value is 0.83>0.7, and the Bartlett's Test of Sphericity is significant. This result means that the data has good structure validity and is suitable for factor analysis. Besides, the competition model comparison methods were used to test the discriminant validity.

Point 5: Can your results be interpreted as the more a citizen in is accustomed to following government directives, the more likely they are to comply if they get exposed to sufficient information, instead of the more information exposure they receive from the media, the more likely they will trust government and comply. I know I'm simplifying here, but am basically asking if this research can transfer to other cultures/populations? if not, the potential difference should be discussed.

Response 5: Thank you for pointing out this issue. According to the results of cross-scenario analysis, more exposure to information that individuals receive from the media, cannot always predict how likely they will trust government and comply. Especially in the NEW-COOP scenario, the relationship between new media exposure and trust in government is totally mediated by knowledge. We modify the sentences in the discussion to emphasize this finding. “However, in the cross-scenario analysis, we found that trust in government can play roles on the relationship between media exposure and cooperation behaviors, while it cannot play roles on the relationship between media exposure and supplies behaviors. Moreover, new media exposure does not have direct effects on trust in government in NME-COOP scenario, which means that simple media exposure may not increase individuals’ trust in government, but need though the growth of knowledge. Therefore, through the effective combination of new and traditional media platforms, various forms of public safety contents for distributing knowledge of public health emergencies, public health emergency management, government activities and events should be communicated to the public in a seamless, comprehensive, and real-time manner in order to improve their trust in government.

Point 6: Finally, in your discussion you state media exposure can increase PHEP, I suggest more details, i.e. number of media you are exposed to, frequency you view media, if exposure to more than one type of media makes a difference, etc.

Response 6: Thank you very much for your suggestions. In this study, we measured media exposure by scoring the frequency. In response to your comments, we added the following information: “Specifically, frequently exposure to both traditional media and new media can directly promote their public health supplies behaviors, while the direct effects are not significant between the relationship between media exposure and their cooperation behaviors, in which way, knowledge and trust in government totally mediate the relationship between media exposure and cooperation behaviors.

Reviewer 2 Report

line 17  Omit how to...nsert enhancing.

line 20   Omit the.

line 52   Continue sentence with and issue warnings."

line 58   Omit such as...insert on.

line 61  Omit so that can...insert to.

Beginning line 63 and following...While behavior is the correct British spelling, consider changing to behavior for that word and its derivatives if this will be directed primarily to American audiences.

Line 78 and following  Traditional media is defined on lines 121-123. It needs to be defined in this section. New media also needs to be defined.

line 84   This prediction seems like a bold assumption from just one example.

lines 98-113   More background is needed. Cost of living data would be helpful. Also, a chart of this information would perhaps be easier to follow.

lines 118-128   This would also lend itself to a chart.

line 129   Insert acquisition of emergency before supplies.

line 131   Add ing to refer...referring.

line 195   Analyzing...American English would be analyzing..

line 221 and following  It seems that this was written by a different author...the use of language is more understandable.

line 240   Define COOP.

line 242   Insert emergency before supply.

line 252   What is meant by "mediate"  and "completely mediate?"

line 316   Omit Taking into account...substitute Because.

line 316   Substitute programs for education.

Overall, a more detailed explanation of media sources available in China would be helpful. Are the authors making an assumption that the audience reading this article understand both the formal and informal lines of communication available in China?

Author Response

Thank you very much for reviewing this paper closely despite your busy schedule. We have carefully review of our manuscript and have tried to address all of your useful feedback and revised our manuscript for greater clarity. Below we have replied to the reviewer’s comments and provided extensive explanations for the changes that were made in the revised manuscript. In addition, the revised manuscript was reviewed by professional English editors. We believe the paper has been greatly improved as a result of your feedback and represents a greater contribution to the literature.

Point 1: line 17 Omit how to...nsert enhancing.

Response 1: Thank you for pointing out this issue. We have carefully reviewed our manuscript and have tried to revisions to improve the overall clarity of the manuscript. In addition, we have carefully checked for typos or grammatical errors throughout the paper.

Point 2: line 20   Omit the.

Response 2: Corrected.

Point 3: line 52   Continue sentence with and issue warnings."

Response 3: In the revised manuscript, we have rephrased these sentences. 

Point 4: line 58   Omit such as...insert on.

Response 4: Corrected.

Point 5: line 61 Omit so that can...insert to.

Response 5: Corrected.

Point 6: Beginning line 63 and following...While behavior is the correct British spelling, consider changing to behavior for that word and its derivatives if this will be directed primarily to American audiences.

Response 6: Modified in American English. Thank you for your suggestions.

Point 7: Line 78 and following Traditional media is defined on lines 121-123. It needs to be defined in this section. New media also needs to be defined.

Response 7: Thank you for your comments. We have defined traditional media and new media in the section “study participants.”

Point 8: line 84   This prediction seems like a bold assumption from just one example.

Response 8: Thank you for your comments. We have removed ambiguous statements in the revised manuscript.

Point 9: lines 98-113   More background is needed. Cost of living data would be helpful. Also, a chart of this information would perhaps be easier to follow.

Response 9: Thank you very much for your suggestions. In response to your comments, we have added a new table to better describe the socio-demographic characteristics. However, in the survey, we did not consider the variable of the cost of living. In future studies, we will try to consider the influence of cost of living in the research of emergency preparedness behaviors.

Point 10: lines 118-128   This would also lend itself to a chart.

Response 10: Thank you for your comment. In the revised manuscript, a new summary table was provided.

Point 11: line 129 Insert acquisition of emergency before supplies.

Response 11: As you suggested, we have modified the sentence.

Point 12: line 131 Add ing to refer...referring.

Response 12: Corrected.

Point 13: line 195 Analyzing...American English would be analyzing.

Response 13: We have corrected this in the revised manuscript.

Point 14: line 221 and following It seems that this was written by a different author...the use of language is more understandable.

Response 14: Thank you for taking a closer look at our paper. We have carefully reviewed our manuscript and tried to improve the clarity and readability of the paper.

Point 15: line 240   Define COOP.

Response 15: Thank you for bringing our attention to this point. As you suggested, we defined COOP (Cooperative behavior) which refers to the public health emergency behavior that requires the collaboration of two or more individuals and agents, e.g., skills training, emergency drill, volunteer activity. Besides, we unified the abbreviations of the variables in the paper.

Point 16: line 242   Insert emergency before supply.

Response 16: As suggested, we have added emergency before supply to clarify the meaning.

Point 17: line 252   What is meant by "mediate” and "completely mediate?"

Response 17: It was used as an emphasis, but it was deleted the word “completely” from the revised manuscript.

Point 18: line 316   Omit Taking into account...substitute Because.

Response 18: corrected

Point 19: line 316   Substitute programs for education.

Response 19: As you suggested, we have corrected this in the revised manuscript.

Point 20: Overall, a more detailed explanation of media sources available in China would be helpful. Are the authors making an assumption that the audience reading this article understand both the formal and informal lines of communication available in China?

Response 20: Thank you for highlighting this issue. We have added one paragraph to explain the media sources available in China in Section “Study Participants.” “In China, after SARS in 2003, more attention had been paid to government information disclosure. Traditional media (e.g., newspaper, magazine, radio, TV) were effectively used by the government to disclose public information, such as government performance and public service information. In 2006, the national government website system was established, which means that information disclosure can be carried out through new media (e.g., Internet, mobile phone customization). In late 2013, the central government of China put Micro.blog and WeChat as equally important new media channels of government information disclosure as government websites.”